# Environmental Impact of Two Plant-Based, Isocaloric and Isoproteic Diets: The Vegan Diet vs. the Mediterranean Diet

**DOI:** 10.3390/ijerph20053797

**Published:** 2023-02-21

**Authors:** Denise Filippin, Anna Rita Sarni, Gianluca Rizzo, Luciana Baroni

**Affiliations:** 1Scientific Society for Vegetarian Nutrition, 30171 Venice, Italy; 2Independent Researcher, Via Venezuela 66, 98121 Messina, Italy

**Keywords:** food system, climate change, life cycle assessment, LCA, environmental impact, sustainable diet, plant-based diets, environmental footprint

## Abstract

Food consumption is one of the major causes of climate change, resource depletion, loss of biodiversity, and other kinds of environmental impact by modern households. According to evidence, a global change in dietary habits could be the single most effective and rapid intervention to reduce anthropic pressure on the planet, especially with respect to climate change. Our study applied Life Cycle Assessment (LCA) to investigate the total environmental impact of two plant-based diets: the Mediterranean and the Vegan diets, according to relevant Italian nutritional recommendations. The two diets share the same macronutrient rates and cover all the nutritional recommendations. Calculations were made on the basis of a theoretical one-week 2000 kcal/day diet. According to our calculations, the Vegan diet showed about 44% less total environmental impact when compared to the Mediterranean diet, despite the fact that the content of animal products of the latter was low (with 10.6% of the total diet calories). This result clearly supports the concept that meat and dairy consumption plays a critical role, above all, in terms of damage to human health and ecosystems. Our study supports the thesis that even a minimal-to-moderate content of animal foods has a consistent impact on the environmental footprint of a diet, and their reduction can elicit significant ecological benefits.

## 1. Introduction

“*Sustainability is the development that meets the needs of the present, without compromising the ability of future generations to meet their own needs*” [1].

The present phase of our planet is called the “*Anthropocene*”, an era in which one single species is altering the Earth’s systems, causing climate change, biodiversity loss, land and water scarcity, and many other environmental issues. People are living well beyond Earth’s means, cumulating an “environmental deficit” that started about 35 years ago [2], which is compromising our sustainability.

For the European Union (EU), progress towards reaching the 17 Sustainable Development Goals (SDGs) of the United Nation’s 2030 Agenda for Sustainable Development, to be achieved by 2030, will require increased efforts in the optimization of food production and distribution, climate change mitigation, and resource preservation [3]. According to FAO/WHO, sustainable diets should provide “adequate, safe, diversified and nutrient rich food for all, which contribute to healthy diets” [3,4,5].

Some specific areas can be influenced by the food production process, i.e., desertification (water scarcity), land degradation and food security. A 2 °C global warming is deemed to increase the risk of food system instability [6].

Data from the Food and Agriculture Organization (United Nations) show that only 29% of the Earth’s surface is covered with land, 71% of which is habitable. As much as 50% of the habitable land is devoted to agriculture, of which 77% is used for animal farming, a land amount that produces only 18% of the total calorie supply [7]. With a projected world population of 9 billion people, the growing meat consumption and the use of bio-based materials and biofuels will cause an estimated increase of 70–110% in agricultural production by 2050 [8] (Figure 1).

The food system is tightly bound to the environment because it relies on it for most of its primary inputs: (a) the consumption of natural resources (water, land, soil, seeds etc.), and (b) the introduction of several residual emissions into the environment, in the form of wasted food, pollutants like pesticides, drugs (e.g., antibiotics) and GHGs, which have an impact also on human health. This interrelationship is clearly complex and multidisciplinary [3].

Diets have been traditionally conceived as factors and strategies interrelated to health and well-being and influence the diet-related incidence of diseases. Diets, nevertheless, relate also to the food system, which has been recognized as a major source of environmental impact, with a close relation to several of the so-called planetary boundaries [9].

In fact, in addition to what the choices on a large scale can contribute, the choices of every single person are also important. In this context, it has been reported that individual dietary choices can help influence sustainability. Specifically, it has been demonstrated that foods of plant origin are more sustainable. Therefore, their proportion in the diet influences the total environmental impact of the diet itself. Omnivorous dietary patterns are known to have a higher impact on the environment than plant-based diets, and the amount of animal foods in the diet appears to be the major determinant of the total impact [10,11].

The original Mediterranean diet, although omnivorous, can be considered, if well planned, a plant-based diet since it emphasizes whole plant foods (vegetables, fruits, nuts, whole grains, and olive oil), despite including small amounts of animal foods (dairy, fish and poultry, and red meat) [12].

On the contrary, in the Vegan diet, all animal foods are totally absent: composition is based on grains, legumes, vegetables, fruits, nuts and oils [13].

Therefore, the only qualitative difference between the two diets is that the animal protein foods of the Mediterranean diet are replaced by protein plant foods in the Vegan diet.

Based on this principle, we aimed to evaluate if and how much a Vegan diet, with a comparable energy-nutritional composition, could represent a real advantage in terms of the total environmental impact compared to the Mediterranean diet.

We used LCA methodology to investigate the total environmental impact of the two plant-based diets by SimaPro^®^ and Ecoinvent^®^, which are the most commonly used LCA software and database [14].

## 2. Materials and Methods

Our aim was to compare two well-planned plant-based diets, the Mediterranean and the Vegan, both healthy and environmentally friendly, to assess how they differ in their environmental impacts even though they share as many similarities as the respective guidelines consent, in terms of nutritional values and gastronomic preparations.

The two diets, which were formulated by a licensed dietician, were similar in terms of the quantity and nutritional composition of the foods consumed and were conceived to minimize “composition biases”: they share the same sources of food types, the same nutrient compositions, the same recipes (where applicable), and the same amount of energy. Differences in the use of non-protein foods were reduced as much as their respective guides suggested. In the Mediterranean diet, we planned only 10.6% of its total calories are derived from animal foods, which puts it under the umbrella of the “plant-based diets”.

The functional units are quantified descriptions of a product’s function, used as the basis to calculate impact assessments. In our study, the functional units consist of all the “ready to eat” food products of two 2000 kcal/day “one-week diets” (Table 1a), each planned according to their respective dietary guides (Mediterranean and Vegan) and carefully developed to minimize unnecessary differences that could bias the final result.

In our study, the Mediterranean diet was planned according to the “new revised MD pyramid representation”, published in 2011 [12], whether the Vegan diet’s planning derived from the Mediterranean “VegPlate” guide [13].

Due to the high-calorie density of animal foods, the quantity, in grams/day, of these products was significantly lower than in an average Western diet. Calorie and nutrient intake counts were obtained using MetaDieta^®^ professional software, using the Italian food database [15] (Table 1b).

Recipes were simplified, avoiding unnecessary steps in their preparations in all calculations. For example, we used “mixed boiled vegetables” for all recipes containing cooked vegetables, “mixed cooked meat” (cow meat, swine meat and chicken meat) for all recipes containing meat and so on. Both diets contain the same amount of cooked grains, pasta and vegetables (both raw and cooked) but differ in the amount of fruit, nuts, oils and protein foods due to the differences in their respective guides.

Life cycle assessment (LCA) is an analytical and systematic methodology that evaluates the environmental footprint of a product or service along its entire life cycle.

We used an internationally recognized method of evaluating environmental impact (LCA): ReCiPe 2016 [16]. At the midpoint level, 18 impact categories were addressed. They were then aggregated into endpoint damage categories. Midpoints included: climate change (human health, terrestrial ecosystem and freshwater ecosystem), stratospheric ozone depletion, ionizing radiation, ozone formation (human health), fine particulate matter formation, ozone formation (terrestrial ecosystems), terrestrial acidification, freshwater eutrophication, marine eutrophication, terrestrial ecotoxicity, freshwater ecotoxicity, marine ecotoxicity, human carcinogenic toxicity, human non-carcinogenic toxicity, land use, mineral resource scarcity, fossil resource scarcity, and water consumption (human health, terrestrial ecosystem and aquatic ecosystem). At the endpoint level, most of these midpoint impact categories are multiplied by damage factors and aggregated into three endpoint categories: human health, ecosystems, and resource [16].

In this study, Software SimaPro^®^ was used for LCA analysis. Given the absence of an Italian national database for inventory, we used the Ecoinvent-3 library, which contains LCI data from various sectors (e.g., energy production, goods transportation, production of chemicals, metal production, fruit and vegetable production etc.). We assessed the impact of the two selected diets based on the “cradle to gate”, or “farm to fork” system boundaries, which includes all the processes involved in the production of our unit (i.e., the one-week diet) up until its consumption. The system boundaries we selected included the following sub-stages: (1) agricultural food production (crops, animal husbandry), (2) transport (global), (3) processing of food products (for the general market), (4) packaging, and (5) consumption, including home preparation. Data for sub-stages 1 to 4 were derived from the Ecoinvent^®^ database. Sub-stage 5 calculations were not provided in this paper but are available on request. Other sub-stages, such as transportation to retailers, waste, food losses and recycling, were excluded. Nevertheless, some downstream emissions, such as those that occur in food processing or preparation (e.g., kitchen gas, water, and electricity used), were included in the calculations. For example, some plant foods require longer cooking times, so their impact has been assessed in the contexts of all food system activities, from production to consumption.

## 3. Results

The results of the Assessment (Life Cycle Impact Assessment/LCIA)—Calculations are shown in Table 2 and Table 3 and Figure 2.

To make easier the comparison of the values, we used the same order of magnitude for each category (same row) in the tables.

In Figure 2, the colors indicate the contribution of the two diets: light green for the Vegan diet, and brown for the Mediterranean diet.

### 3.1. Characterization

Table 2a and Figure 2a provide a closer look at the contributions of the two diets to various categories of Impact (midpoint characterization factors).

In this step, all substances are multiplied by characterization factors (CF), which quantifies how much impact a single unit of a product has in the various categories of environmental impact. In Figure 2a, all impact scores are displayed on a 100% scale.

### 3.2. Damage Assessment

This step (endpoints) aggregates a number of impact category indicators into a Damage category. At this stage of the calculation (Table 2b, Figure 2b), the difference in the impact of the two diets for the three Damage categories is evident: the Vegan diet scores almost half of the impact of the Mediterranean diet with respect to the human health and the ecosystems endpoints. Also, in Figure 2b, all impact scores are displayed on a 100% scale.

### 3.3. Normalization

In this step (Table 2c, Figure 2c), the impact is compared to a reference value, termed “normalization reference”. It is a major factor in the aggregation process and facilitates comparisons, comprehension, communication, and decision-making. The results of this step confirm the effects on ecosystems and human health of the previous phase.

### 3.4. Weighted Average to Obtain a Single Score

Weighting results are reported in Figure 2d,e, and Table 3.

Despite the small difference in the amount of animal products (10.6% of the total calories), the Vegan diet’s overall impact (Single Score) is 43.88% lower than the Mediterranean diet’s impact.

Table 3 also provides the detailed values of the Impact categories’ Points, which quantify the contributions of the two diets to the categories of Impact (midpoint characterization factors): the higher the value, the higher the impact. The effects of the various Impact categories will be commented on in Section 4, the Discussion.

The differences between the two diets are also expressed in percentages (Table 3, last column).

For a better understanding of the single contribution of each food group (Process Contribution), we propose some details of their impact, calculated with the Single Score ReCiPe 2016 Endpoint H method [16]. Data are presented in Appendix A.

The impact of the two diets on CO_2_-equivalent emission has also been calculated by using the evaluation method developed by the Intergovernmental Panel on Climate Change (IPCC), available on SimaPro^®^, and is shown in Appendix B.

In Appendix C, we propose two examples of comparison between animal and analogous protein plant foods.

## 4. Discussion

In addition to human health, over the last few years, researchers have begun investigating dietary strategies as a means of reducing environmental impacts due to the food system. For instance, the food system was estimated to contribute between 19% and 29% of global greenhouse gas (GHG) emissions and to account for approximately 70% of freshwater use globally [6,9,17].

In order to achieve sustainable diets, which must also be healthy, a public strategy should focus on improving energy balance and dietary changes toward predominantly plant-based diets that are consistent with healthy eating guidelines [18].

Reducing our reliance on animal foods is widely acknowledged as one of the most effective ways—on the individual level—to reduce our environmental impact on climate change, i.e., GHG production, and on other aspects like land use, pollutant emissions etc. [17,19,20,21].

The food system represents the primary driver of land use [22]: the land is tightly interrelated with climate change and, consequently, GHG emissions. Methane and nitrous oxide, which are potent GHGs produced by livestock, are short-lived if compared to CO_2_ itself. A phaseout of livestock production, and the consequent land restoration, even in the absence of any other emission reductions, would translate into a first rapid reduction of GHGs, due to the decay of the two gases [23].

We have previously used LCA methodology to compare the environmental impact of Lacto-ovo-vegetarian, Vegan and Omnivorous balanced diets, showing that the lower the animal food contribution of the diet, the lower the impact was [10,11,24].

To our knowledge, no study has so far compared the total environmental impact of the Mediterranean diet with that of the Vegan diet. The available studies either did not compare the two diets contextually or compared only some of the impacts but not the total impact [25,26,27,28,29,30,31,32,33,34,35].

The original Mediterranean diet is a balanced diet meeting nutritional recommendations. It is based mainly on plant foods, seasonal and locally available, whose consumption goes hand in hand with their production and the social and cultural factors that make this diet so typical, to the point of being recognized as an “Intangible Cultural Heritage of Humanity” [36,37].

However, many factors, among which mainly globalization and the advent of modern food production techniques, with the change of traditional habits, are leading to a progressive reduction of the population’s adherence to the Mediterranean diet [38].

Although the Mediterranean diet is still considered culturally acceptable, cheap, and healthy [39,40], the Vegan diet is becoming utmost popular.

Foods composing a Vegan diet are very similar to those of the Mediterranean tradition. A well-planned Vegan diet is considered nutritionally adequate and healthy [41], and in a study that compared the cost of different diets, the results presented the Vegan diet to be the cheaper one [25].

The 2015 (Updated in 2021) Dietary Guidelines Scientific Advisory Committee states that “*a dietary pattern that is higher in plant-based foods, such as vegetables, fruits, whole grains, legumes, nuts, and seeds, and lower in animal-based foods is more health promoting and associated with less environmental impact (GHGE and energy, land, and water use) than the current average US diet*” [42].

So, what can be the difference in the total environmental impact of two similar plant-based diets, which respectively greatly limit or eliminate animal foods?

Although the two diets are very similar from a nutritional point of view, the analysis of their respective environmental impact has highlighted important differences, which we discuss below.

Our comparison of the Impact categories of the Mediterranean vs. Vegan diet showed many differences favoring the Vegan one: Human non-carcinogenic toxicity (−71.95%), Land use (−48.78%), Terrestrial acidification (−31.04%), Ozone formation (mean −25.62%), Stratospheric ozone depletion (−23.40%), Fine particulate formation (−24.17%), Global warming (mean −23.2%). Lesser differences favoring the Vegan diet were present for Freshwater eutrophication and ecotoxicity, Human carcinogenic toxicity, Mineral resource scarcity and Fossil resource scarcity (mean −4.54% [from −6.03% to −2.9%]).

The remaining Impact categories were instead in favor of the Mediterranean diet, but except for Ionizing radiation (86.55%), the other ones elicited low differences: Terrestrial and marine ecotoxicity (2.24% and 0.98%, respectively, mean 1.61%) and Water consumption (mean 5.28% [from 3.13% to 7.16%]).

However, a more careful analysis of Table 3 and Figure 2d highlighted that in terms of absolute values, the Impact categories with the highest importance are 4 (the ones highlighted in Bold Italics font): GWHH: Global warming, Human health; FPMF: Fine particulate matter formation; HNCT: Human non-carcinogenic toxicity; LU: Land use. Their contribution to the total impact appeared preponderant compared to the other Impact categories, and the differences between the two diets were always in favor of the Vegan diet.

Moreover, once the Impact categories were aggregated by Damage categories, the Vegan diet was favored for all Damage categories, and its total impact was lower than the Mediterranean diet. LCA calculations (with ReCiPe 2016 Endpoint H [16]) showed that, despite a low difference in the protein foods (representing 10.6% of total kcalories), the total environmental impact of the Vegan diet was 43.88% lower than the Mediterranean diet’s impact, which means that the Mediterranean diet’s impact was 78.18% higher than the Vegan diet’s impact.

This finding confirms the validity of applying the LCA analysis to all the Impact categories to obtain an assessment more in line with real life: a single or few Impact categories may not reflect the entity of the Damage categories and of the total environmental impact of the diet or could even reverse the conclusions.

Our calculations showed that the 10.6% of calories derived from animal products were responsible for about half (47%) of the global impact of the Mediterranean diet, with meat showing the largest contribution (around 30%), despite the minimum amount included (60 g/week). See Appendix A for details.

Regarding protein sources, in our calculations, legumes and seitan had, respectively, a total impact at the Single Score level of about 84% and 32% lower than mixed meat, and soy milk’s total impact was 79% lower than cow’s milk’s one (data are shown in Appendix C).

Considering the climate emergency, we also launched an LCA calculation based on IPCC 2013 GWP (100a) method, available in SimaPro^®^, to test the environmental effect of the two plant-based diets on GHG emission. According to our results, the Vegan diet impact was 78.7% of that of the Mediterranean diet (Appendix B, Figure A4).

There is enough evidence that plant-based diets are both adequate and protective against the most widespread chronic diseases in the developed world [41]. However, food systems should also be economically viable and improve food security, prevent malnutrition and reduce environmental degradation [5,43].

Thanks to LCA analysis, we highlighted how, among the various impact categories especially affected by the two diets, one consistently emerged: animal-derived foods represent a significant burden for the Earth’s soil. Soil scarcity is an insufficiently discussed emergency, given that the Earth’s surface is still free from human activities represents a fundamental factor for our survival and for ecological balance. Land scarcity also threatens local food security and biodiversity. In this scenario, the food system is the primary driver of land use, and land scarcity is the primary driver of zoonotic spillovers [44].

Food systems need to deal with human health, national economy and culture, but also address climate change mitigation, tackle the depletion of natural resources, and possibly, not forget workers’ human rights (equity and fair trade).

Although it is claimed that a diet with a moderate amount of animal foods has only a modestly higher impact on the environment, with respect to a plant-only diet, in our study, we demonstrated that even modest consumption of animal products had a consistent impact on critical environmental aspects.

According to these data, animal-based food production represents a significant burden for the planet. Given that the average diet is much higher in animal products than the Mediterranean diet we used for comparison, the consequences can be unpredictable.

Food policies should be planned by a multidisciplinary task force, which includes collaboration among scholars and stakeholders from multiple disciplines and sectors.

## 5. Limitations

The environmental impact of food production is region-specific, while we used global market standards. Therefore, there can be relevant differences in environmental impacts when referring to regional or local productions, depending on the origin, quality, distance from the consumption site, traditional processing, price etc. Thus, our findings cannot be directly transferred to a region-specific environmental impact system.

Data used for our calculations were derived from the Ecoinvent^®^ database, which is continuously updated and offers detailed uncertainty characterizations for most energy and material flows in its lifecycle inventory data.

Worth noting when interpreting midpoints results: according to ISO 14044 [45], in LCA calculations, water consumption is not a direct expression of how much water is used in the process. In LCA, a product or service is evaluated for the water impact throughout its entire life cycle, but only the “blue water” is counted. Green water is not considered in LCA, while grey water is partly assessed in a few impact categories. On the other hand, “water footprint” evaluates water based on volumetric use, and this method quantifies and maps green, blue and grey water [46]. There’s an ongoing and heated debate about whether the water footprint should be a volumetric or an impact-based indicator [47]. So, compared to other impact categories, our water consumption results are especially uncertain since the amount of water is strictly linked to local conditions such as rainfall, irrigation, evapotranspiration and pedoclimatic elements [48].

In simple terms, sustainable diets are context specific. Important factors like the local climate, the physical properties of soil and land, water availability, and many others, including the diversity of agricultural production systems and local environmental settings, as well as local culture, should be taken into consideration in the decision-making process of sustainable diets and sustainable food systems.

Nevertheless, despite the uncertainty and variability inherent in these complicated systems, this simple underlying trend provides relatively high confidence in the direction of the conclusions.

## 6. Conclusions

Diet has an impact on both health and the ecosystem. In our work, we have compared two sustainable diets with very similar nutrient compositions but with substantial differences in their total environmental impacts. The replacement of a small calorie quota (10.6%) represented by animal foods with plant foods showed significant improvement in the total environmental impact, especially for ecosystems and human health.

This suggests that the more plant-based the diet is, the less it will impact the environment. This information is noteworthy in light of how many countries show a diet rich in animal foods and how much this represents a global risk to sustainability.

However, while the health consequences are already known, there is still little attention on the environmental outcomes, given how even small amounts of animal food can make a difference.

## Figures and Tables

**Figure 1 ijerph-20-03797-f001:**
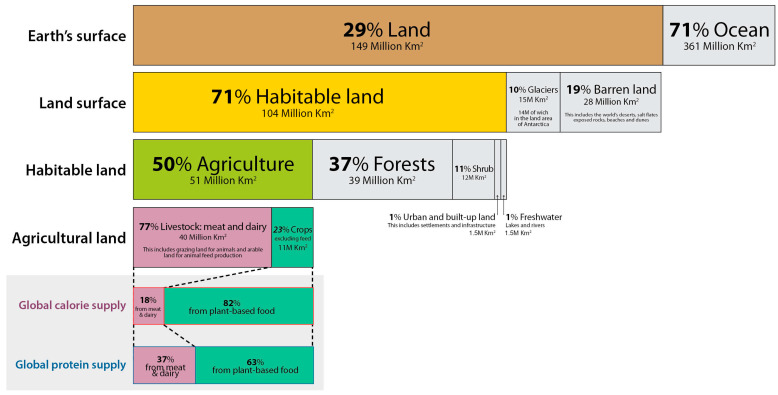
Land Use (Open-source under the CC-BY License) [7].

**Figure 2 ijerph-20-03797-f002:**
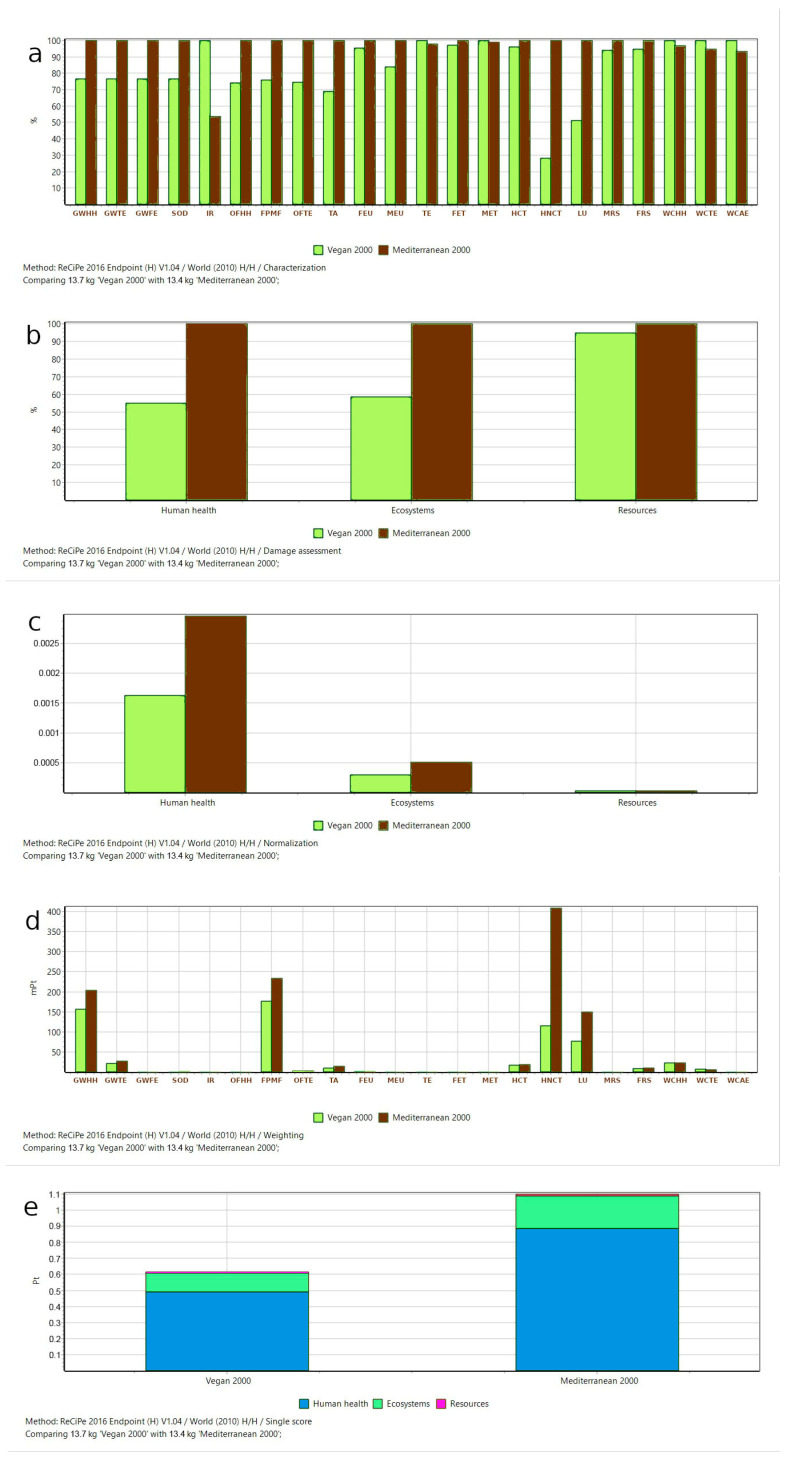
Comparison between Vegan diet and Mediterranean diet (2000 kcal): (**a**) Characterization; (**b**) Damage Assessment; (**c**) Normalization; (**d**) Weighting; (**e**) Aggregated Weighted Average (Single Score). GWHH: Global warming, Human health; GWTE: Global warming, Terrestrial ecosystems; GWFE: Global warming, Freshwater ecosystems; SOD: Stratospheric ozone depletion; IR: Ionizing radiation; OFHH: Ozone formation, Human health; FPMF: Fine particulate matter formation; OFTE: Ozone formation, Terrestrial ecosystems; TA: Terrestrial acidification; FEU: Freshwater eutrophication; MEU: Marine eutrophication; TE: Terrestrial ecotoxicity; FET: Freshwater ecotoxicity; MET: Marine ecotoxicity; HCT: Human carcinogenic toxicity; HNCT: Human non-carcinogenic toxicity; LU: Land use; MRS: Mineral resource scarcity; FRS: Fossil resource scarcity; WCHH: Water consumption, Human health; WCTE: Water consumption, Terrestrial ecosystem; WCAE: Water consumption, Aquatic ecosystems.

**Table 1 ijerph-20-03797-t001:** Vegan vs. Mediterranean diet foods.

**a. Composition of the Two One-Week Diets.**						
**Vegan**			**Mediterranean**			
**Food**	**Amount**	**Unit**	**Food**		**Amount**	**Unit**
Mixed grains (cooked)	1.12	kg	Mixed grains (cooked)		1.12	kg
Rice (cooked)	0.72	kg	Rice (cooked)		0.72	kg
Pasta (cooked)	1.68	kg	Pasta (cooked)		1.68	kg
Bread	0.66	kg	Bread		0.66	kg
Olive oil	0.08	kg	Olive oil		0.14	kg
Mixed legumes (cooked)	0.88	kg	Mixed legumes (cooked)		0.40	kg
Mixed fruit	2.63	kg	Mixed fruit		2.10	kg
Vegetables (raw and cooked)	4.20	kg	Vegetables (raw and cooked)		4.20	kg
Mixed nuts	0.42	kg	Mixed nuts		0.40	kg
Sunflower oil	0.06	kg	Egg (cooked)		0.12	kg
Soy dessert, plain, refrigerated	0.25	kg	Chicken (cooked)		0.12	kg
Soy drink, plain, fortified with calcium	0.80	kg	Cheese		0.21	kg
Seitan	0.06	kg	Fish (cooked)		0.12	kg
Tofu	0.16	kg	Red meat (cooked)		0.06	kg
			Skimmed milk		1.40	kg
**b. Daily Average Nutritional Characteristics of the Two One-Week Diets.**						
	**Vegan**			**Mediterranean ***		
Energy (kcal)	2016.57			2018.57		
Carbohydrates (%)	52.93			49.76		
Proteins (%)	16.48			17.81		
Fats (%)	30.55			32.46		
Fiber g (total/1000 kcal)	24.14			21.09		
Iron (mg)	22.52			19.27		
Calcium (mg)	851.94			853.48		
Zinc (mg)	12.71			12.98		

* Total kcalories from animal products in a week 1500; kcalories from animal products/per day 214.28.

**Table 2 ijerph-20-03797-t002:** Life Cycle Assessment Calculation (LCIA), proposed for steps. (DALY: disability-adjusted life years; species.yr: loss of species during a year; USD2013: US Dollars).

**a. Characterization**				
**Impact Category**	**Unit**	**Vegan**	**Mediterranean**	**Δ Veg-Med**
Global warming, Human health	DALY	1.24 × 10^−5^	1.62 × 10^−5^	−0.38 × 10^−5^
Global warming, Terrestrial ecosystems	species.yr	3.75 × 10^−8^	4.88 × 10^−8^	−1.13 × 10^−8^
Global warming, Freshwater ecosystems	species.yr	1.02 × 10^−12^	1.33 × 10^−12^	−0.31 × 10^−12^
Stratospheric ozone depletion	DALY	4.77 × 10^−8^	6.23 × 10^−8^	−1.46 × 10^−8^
Ionizing radiation	DALY	6.13 × 10^−9^	3.28 × 10^−9^	2.84 × 10^−9^
Ozone formation, Human health	DALY	3.08 × 10^−8^	4.14 × 10^−8^	−1.06 × 10^−8^
Fine particulate matter formation	DALY	1.40 × 10^−5^	1.85 × 10^−5^	−0.45 × 10^−5^
Ozone formation, Terrestrial ecosystems	species.yr	4.45 × 10^−9^	5.98 × 10^−9^	−1.53 × 10^−9^
Terrestrial acidification	species.yr	1.75 × 10^−8^	2.54 × 10^−8^	−0.79 × 10^−8^
Freshwater eutrophication	species.yr	2.99 × 10^−9^	3.14 × 10^−9^	−0.15 × 10^−9^
Marine eutrophication	species.yr	2.97 × 10^−11^	3.54 × 10^−11^	−0.06 × 10^−11^
Terrestrial ecotoxicity	species.yr	5.56 × 10^−10^	5.43 × 10^−10^	0.12 × 10^−10^
Freshwater ecotoxicity	species.yr	5.22 × 10^−10^	5.37 × 10^−10^	−0.16 × 10^−10^
Marine ecotoxicity	species.yr	8.87 × 10^−11^	8.78 × 10^−11^	0.09 × 10^−11^
Human carcinogenic toxicity	DALY	1.36 × 10^−6^	1.42 × 10^−6^	−0.05 × 10^−6^
Human non-carcinogenic toxicity	DALY	9.08 × 10^−6^	32.36 × 10^−6^	−23.28 × 10^−6^
Land use	species.yr	1.37 × 10^−7^	2.67 × 10^−7^	−1.30 × 10^−7^
Mineral resource scarcity	USD2013	1.64 × 10^−2^	1.75 × 10^−2^	−0.11 × 10^−2^
Fossil resource scarcity	USD2013	8.29 × 10^−1^	8.76 × 10^−1^	−0.47 × 10^−1^
Water consumption, Human health	DALY	1.84 × 10^−6^	1.79 × 10^−6^	0.06 × 10^−6^
Water consumption, Terrestrial ecosystem	species.yr	1.18 × 10^−8^	1.12 × 10^−8^	0.06 × 10^−8^
Water consumption, Aquatic ecosystems	species.yr	4.17 × 10^−12^	3.89 × 10^−12^	0.28 × 10^−12^
**b. Damage Assessment**				
**Damage Category**		**Vegan**	**Mediterranean**	**Δ Veg-Med**
Human Health	DALY	3.88 × 10^−5^	7.03 × 10^−5^	−3.15 × 10^−5^
Ecosystems	Species.yr	2.12 × 10^−7^	3.63 × 10^−7^	−1.50 × 10^−7^
Resources	USD2013	0.8457	0.8933	−0.0476
**c. Normalization**				
**Damage Category**	**Unit**	**Vegan**	**Mediterranean**	**Δ Veg-Med**
Human Health	-	1.63 × 10^−3^	2.96 × 10^−3^	−1.33 × 10^−3^
Ecosystems	-	2.96 × 10^−4^	5.06 × 10^−4^	−2.10 × 10^−4^
Resources	-	3.02 × 10^−5^	3.19 × 10^−5^	−0.17 × 10^−5^

**Table 3 ijerph-20-03797-t003:** Total Environmental Impact of Vegan and Mediterranean diets: Impact Categories and Damage Categories. Aggregated Weighted Average: total environmental load expressed as a Single Score (mPt = milliPoints).

Impact Category	Unit	Vegan	Mediterranean	Δ Veg-Med	%
** *Global warming, Human health* **	** *Pt* **	** *1.57 × 10^−1^* **	** *2.04 × 10^−1^* **	** *−0.47 × 10^−1^* **	** *−23.21* **
Global warming, Terrestrial ecosystem	Pt	2.09 × 10^−2^	2.72 × 10^−2^	−0.63 × 10^−2^	**−23.20**
Global warming, Freshwater ecosystem	Pt	5.71 × 10^−7^	7.44 × 10^−7^	−1.73 × 10^−7^	**−23.20**
Stratospheric ozone depletion	Pt	6.03 × 10^−4^	7.87 × 10^−4^	−1.84 × 10^−4^	**−23.40**
Ionizing radiation	Pt	7.74 × 10^−5^	4.15 × 10^−5^	3.59 × 10^−5^	**86.55**
Ozone formation, Human health	Pt	3.89 × 10^−4^	5.23 × 10^−4^	−1.34 × 10^−4^	**−25.69**
** *Fine particulate matter formation* **	** *Pt* **	** *1.77 × 10^−1^* **	** *2.33 × 10^−1^* **	** *−0.56 × 10^−1^* **	** *−24.17* **
Ozone formation, Terrestrial ecosystem	Pt	2.48 × 10^−3^	3.34 × 10^−3^	−0.85 × 10^−3^	**−25.55**
Terrestrial acidification	Pt	9.79 × 10^−3^	14.19 × 10^−3^	−4.40 × 10^−3^	**−31.04**
Freshwater eutrophication	Pt	1.67 × 10^−3^	1.75 × 10^−3^	−0.08 × 10^−3^	**−4.70**
Marine eutrophication	Pt	1.66 × 10^−5^	1.97 × 10^−5^	−0.32 × 10^−5^	**−15.95**
Terrestrial ecotoxicity	Pt	3.10 × 10^−4^	3.03 × 10^−4^	0.07 × 10^−4^	**2.24**
Freshwater ecotoxicity	Pt	2.91 × 10^−4^	3.00 × 10^−4^	−0.09 × 10^−4^	**−2.90**
Marine ecotoxicity	Pt	4.95 × 10^−5^	4.90 × 10^−5^	0.05 × 10^−5^	**0.98**
Human carcinogenic toxicity	Pt	1.72 × 10^−2^	1.79 × 10^−2^	−0.07 × 10^−2^	**−3.74**
** *Human non-carcinogenic toxicity* **	** *Pt* **	** *1.15 × 10^−1^* **	** *4.09 × 10^−1^* **	** *−2.94 × 10^−1^* **	** *−71.95* **
** *Land use* **	** *Pt* **	** *0.76 × 10^−1^* **	** *1.49 × 10^−1^* **	** *−0.73 × 10^−1^* **	** *−48.78* **
Mineral resource scarcity	Pt	1.76 × 10^−4^	1.87 × 10^−4^	−0.11 × 10^−4^	**−6.03**
Fossil resource scarcity	Pt	8.88 × 10^−3^	9.38 × 10^−3^	−0.50 × 10^−3^	**−5.32**
Water consumption, Human health	Pt	2.33 × 10^−2^	2.26 × 10^−2^	0.07 × 10^−2^	**3.13**
Water consumption, Terrestrial ecosystem	Pt	6.58 × 10^−3^	6.23 × 10^−3^	0.35 × 10^−3^	**5.55**
Water consumption, Aquatic ecosystem	Pt	2.33 × 10^−6^	2.17 × 10^−6^	0.16 × 10^−6^	**7.16**
**Damage Category**					
Human health	Pt	4.90 × 10^−1^	8.88 × 10^−1^	−3.98 × 10^−1^	**−44.83**
Ecosystems	Pt	1.18 × 10^−1^	2.02 × 10^−1^	−0.84 × 10^−1^	**−41.50**
Resources	Pt	9.06 × 10^−3^	9.57 × 10^−3^	−0.51 × 10^−3^	**−5.34**
**Total**	Pt	6.17 × 10^−1^	11.00 × 10^−1^	−4.83 × 10^−1^	**−43.88**

Note: Impact categories with the highest importance are in Bold Italics.

## Data Availability

All the data presented in this study are available on request from the corresponding author.

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
