# Peer review of "Environmental Impact of Two Plant-Based, Isocaloric and Isoproteic Diets: The Vegan Diet vs. the Mediterranean Diet"

_ijerph, 2023, doi:10.3390/ijerph20053797_

Round 1
Reviewer 1 Report
the manuscript "Environmental Impact of Two Plant-based, Isocaloric and Isoproteic Diets: The Vegan Diet vs. the Mediterranean Diet" deals with the assessment of two different diet models in terms of environmental impact.
The topic is interesting. However, the presentation of the methods and results, as well as their discussion, is very inconsistent.
1. Introduction should be lead the reader to understand the specific aim of the article. However, in this manuscript seems a collage of concepts without a logical framework from general topic to the specific aim.
2. Materials and methods: different points lack. For example: where did the authors retrieve diet composition data from? Chemical composition of the different food items? For what country? Morever, no uncertainty analysis has been provided.
About LCA, it is not necessary to describe in detail the method per se but how the authors used it in their study.
In this version, there is no possibility to analyse methods for a possibile replication
3. Results. In LCA, the presentation of the mid point results is mandatory (e.g. in terms of kg CO2-eq for global warming potential), before presenting normalized results and/or endpoint results.
4. what is the novelty of this manuscript? There are many studies about the environmental footprint of vegan and mediterranean diets
5. Presentation: I suggest reviewing the text in order to enhance the logical flow of the sentences.
Author Response
Please, see the attached file

Reviewer 2 Report
It is unquestionably that a diet containing even minimal animal ingredients presents significant environmental impacts. Moreover, it is equally true that a comparison between diets (i.e., the vegan and Mediterranean) should be objective.
As such, in the Results section (i.e, paragraph 3, lines 223 and following), the authors address the negative effects of the DM on some impact categories (i.e., global warming, ozone, formation and depletion, land use, terrestrial acidification, particulate formation, marine eutrophication and human non-carcinogenic toxicity). In this context, it should be grateful that the authors complete the analysis by addressing how the vegan diet presents greater impacts than the DM in terms of WHCN, WCTE, WCAE, IR, TE & MET.
Finally, considering how over time the DM has reached scientific consensus in terms of human health, it would be grateful that the authors address also in their manuscript some of its peculiarities, such as:
. DM encourages the consumption of fresh seasonal products;
. DM presents economic sustainability;
. DM affects the socio-cultural aspects with particular attention to its effects on the local economy.
Author Response
Please, see attached file

Round 2
Reviewer 2 Report
The work has been sufficiently improved. The authors have considered the proposed suggestions, therefore it can be accepted in its current form
Author Response
please see attached file, thank you
